# Low-Frequency Noise Modeling of 4H-SiC Metal-Oxide-Semiconductor Field-Effect Transistors

Yuan Liu [1] , Weijie Ye [2], Xiaoming Xiong [1] and Wanling Deng [2],*

1   School of Integrated Circuits, Guangdong University of Technology, Guangzhou 510006, China
2   Key Laboratory of New Semiconductors, Devices of Guangdong Higher Education Institutes, Department of Electronic Engineering, Jinan University, Guangzhou 510630, China
*   Correspondence: dengwl@jnu.edu.cn

**Abstract:** 4H-silicon carbide metal-oxide-semiconductor field-effect transistors (4H-SiC MOSFETs) show $1/f$ low-frequency noise behavior. In this paper, this can be explained by the combination of the mobility fluctuation ($\Delta\mu$) and the carrier number fluctuation ($\Delta N$) theories. The $\Delta\mu$ theory believes that LFN is generated by the bulk defects, while the $\Delta N$ theory holds that LFN originates from the extraordinarily high oxide traps. For 4H-SiC MOSFETs, significant subthreshold noise will appear when only the $\Delta N$ theory attempts to model LFN in the subthreshold region. Therefore, we account for the high density of bulk defects ($\Delta\mu$ theory) and characterize the subthreshold noise. The theoretical model allows us to determine the bulk density of the trap states. The proposed LFN model is applicable to SiC MOSFETs and accurately describes the noise experimental data over a wide range of operation regions.

**Keywords:** 4H-SiC; MOSFETs; significant subthreshold noise; LFN model





## 1. Introduction

Recently, silicon carbide has become one of the most popular semiconductor materials due to its excellent performance under high pressure, high temperature, high frequency, and high power [1–5]. Metal-oxide-semiconductor field-effect transistors (MOSFETs) fabricated from 4H-silicon carbide (4H-SiC) have also been widely studied in the field of power electronics, and have broad application prospects for their low on-state resistance, high switching speed, good gate insulation, and anti-radiation properties [6–8]. Nevertheless, SiC MOSFETs may still be affected by extreme working conditions leading to degradation [9,10]. Thus, it is necessary to investigate the reliability of SiC power devices. The low-frequency noise (LFN) behavior of semiconductor devices has an important relationship with their reliability [11–13]. Compared with other methods that require the evaluation of the quality and stability of devices under high pressure or high temperature, LFN, a non-destructive and sensitive characterization method of electrical and physical properties [14,15], needs to be further investigated.

In general, there are two classic mainstream theories to explain the LFN behavior of monocrystalline silicon (c-Si) MOSFETs, which are the carrier number fluctuation ($\Delta N$) theory proposed by McWhorter [16] and the mobility fluctuation ($\Delta\mu$) theory proposed by Hooge [17]. Until now, the LFN behavior of SiC MOSFETs has often been explained through the $\Delta N$ mechanism [9,10,18]. It is believed that the oxide-trap-induced carrier number fluctuation prevails and becomes the main noise source. However, in our study, we found that the LFN experimental data of SiC MOSFETs deviated from the noise results predicted by using the $\Delta N$ model. Notably, in the subthreshold region, the noise levels are greater than those in crystalline MOSFETs. The $1/f$ noise level is strongly dependent on the fabrication process, such as interface and active layer qualities. Similar to polysilicon thin film transistors (poly-Si TFTs) [19,20], SiC devices have a high bulk trap state density of $10^{19}$ cm$^{-3}$ eV$^{-1}$ [1]. This further illustrates that only the fluctuations of the interfacial

oxide charge, due to the dynamic trapping and detrapping of free carriers into slow oxide traps, cannot interpret the LFN behavior of SiC MOSFETs.

The aim of the article is to give a complete model for the low-frequency noise characterization of SiC MOSFETs with physical meaning. The model comprehensively accounts for the mechanisms of bulk mobility and carrier number fluctuations. It can accurately describe the high level of low-frequency ($1/f$) noise characteristics in the subthreshold region and fit well with the LFN measured results in the above-threshold region. Furthermore, we use an analytical method to determine the bulk density of states. Finally, the model can be successfully applied to different SiC MOSFETs, which is important for the LFN study of SiC devices.

## 2. Derivation of Noise Model

Low-frequency noise is considered as a powerful tool to evaluate the reliability of semiconductor devices. Much research about noise characteristics has shown that $1/f$ noise (that is, flicker noise) is often the main noise source of LFN [21]. Therefore, the $1/f$ noise spectrum has become an important indicator for evaluating the quality and stability of devices. The major difficulty in noise modeling is to accurately understand the noise source. At present, there are two different explanations for the $1/f$ noise source, i.e., the origins from the interface traps or bulk defects of the devices, which also correspond to the $\Delta N$ and $\Delta \mu$ models, respectively.

The $\Delta N$ model [16] demonstrates that the $1/f$ noise is essentially a surface effect. In other words, the carriers in the active layer of the device can interact with the trapped charges in the oxide layer through the tunneling effect, resulting in the carrier number fluctuation and further leading to LFN. According to the $\Delta N$ model, the normalized drain current noise power spectral density (PSD) can be expressed as

$$\frac{S_{Ids,abv}}{I_{ds}^2} = \left(\frac{g_m}{I_{ds}}\right)^2 S_{Vfb},$$ (1)

where $g_m$ is the device transconductance. $S_{Vfb}$ represents the PSD of flat-band voltage ($V_{fb}$), which can be given by [22]

$$S_{Vfb} = \frac{q^2 kT \lambda N_{ox}}{W L_g C_{ox}^2 f},$$ (2)

where $q$ is the amount of charge; $k$ represents the Boltzmann's constant and $T$ is the temperature value; $\lambda$ refers to the tunnel attenuation coefficient (0.1 nm for SiO$_2$); $N_{ox}$ is the oxide trap density in the gate insulator; $W$ and $L_g$ are the width and length of the gate as shown in Figure 1; it can be seen from Equation (2) that when $L_g$ increases, $S_{Vfb}$ will decrease, and then the noise PSD in Equation (1) will reduce; $C_{ox}$ is the gate oxide layer capacitance per unit area; and $f$ is the device frequency.

However, the $\Delta N$ model is only suitable for describing the noise characteristics of SiC MOSFETs working in the above-threshold region [23], since the bulk trap state density of SiC MOSFETs (about $10^{19}$ cm$^{-3}$ eV$^{-1}$) is much higher than that of conventional c-Si MOSFETs and leads to lower effective channel mobility [1]. Therefore, when the device operates in the subthreshold region with low drain current, the larger channel depth and higher bulk defects will play an important role in the mobility fluctuation phenomenon. It is necessary to consider the contribution of bulk defects to the LFN in the subthreshold region. As a result, the combination of $\Delta \mu$ and $\Delta N$ models is applied to explain the noise behavior in the subthreshold region.

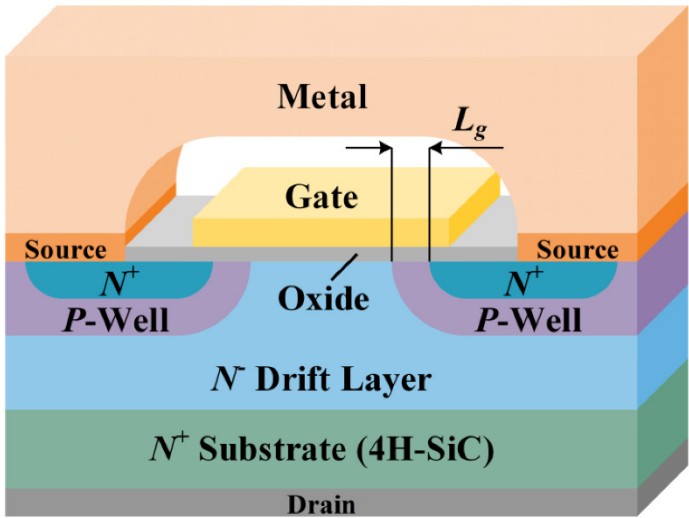

**Figure 1.** Schematic cross-sectional view of a 4H-SiC VDMOSFET elementary cell.

Contrary to the $\Delta N$ model, the $\Delta \mu$ model [17] believes that $1/f$ noise is a bulk effect, which comes from the bulk mobility fluctuation caused by lattice scattering. It is mainly related to the bulk carrier-phonon scattering of the devices, and depends on the channel material and fabrication technology. According to the $\Delta \mu$ model, the normalized drain current noise PSD can be written as

$$\frac{S_{Ids,H}}{I_{ds}^2} = \frac{\alpha_H}{Nf},\tag{3}$$

where $\alpha_H$ is the Hooge empirical constant, which is used to characterize the quality of the channel material and the pros and cons of the interface treatment process. The smaller the value of $\alpha_H$, the better the quality of the device material, and the smaller the amplitude of the LFN caused by this. $N$ represents the total number of free carriers in the channel, and can be calculated by [24]

$$N = \frac{C_{ox}WL_g}{q}\left(V_{gs} - V_{th}\right),\tag{4}$$

where $V_{gs}$ is the gate voltage, $V_{th}$ is the threshold voltage, and the difference between them is the overdrive voltage. By substituting Equation (4) into Equation (3), a more specific $\Delta \mu$ model can be obtained

$$\frac{S_{Ids,H}}{I_{ds}^2} = \frac{q}{C_{ox}WL_g}\frac{\alpha_H}{f}\frac{1}{V_{gs} - V_{th}}.\tag{5}$$

Like the $\Delta N$ model, the noise PSD described by the $\Delta \mu$ model will also reduce as $L_g$ increases. Moreover, for MOSFETs, the drain current in the subthreshold region can be expressed as [25]

$$I_{ds} = qn_s\mu_{eff}WV_{ds}/L_g,\tag{6}$$

where $\mu_{eff}$ is the effective mobility, and $V_{ds}$ is the drain voltage. Parameter $n_s$ is the unified electron sheet charge density per unit area, i.e.,

$$n_s = \frac{2C_{ox}\eta kT}{q^2}\ln\left[1 + 0.5\exp\left(\frac{V_{gs} - V_{th}}{\eta kT/q}\right)\right],\tag{7}$$

where $\eta$ is the subthreshold ideality factor and it is an important parameter characterizing the subthreshold regime. From Equations (6) and (7), the subthreshold drain current expression can be acquired

$$I_{ds} = \frac{2\mu_{eff}C_{ox}\eta kTWV_{ds}}{qL_g}\ln\left[1 + 0.5\exp\left(\frac{V_{gs} - V_{th}}{\eta kT/q}\right)\right].\tag{8}$$

Combining Equations (5) and (8), we obtain

$$\frac{S_{Ids,H}}{I_{ds}^2} = \frac{q^2 \alpha_H / \left( C_{ox} W L_g f \eta kT \right)}{\ln\left\{ 2\left[ \exp\left( \frac{I_{ds} q L_g}{2\mu_{eff} C_{ox} \eta kTWV_{ds}} \right) - 1 \right] \right\}}. \tag{9}$$

The above equation is a Hooge noise model in the subthreshold region that accounts for the bulk mobility fluctuation contributed by device bulk defects, the extraction of which will be discussed in the next section. Since this model is only applicable to the subthreshold regime, it can be further limited by a smooth function as

$$\frac{S_{Ids,sub}}{I_{ds}^2} = \frac{b}{\exp[a(I_{th} - I_{ds})] + 1} + \frac{S_{Ids,H} / I_{ds}^2}{\exp[a(I_{ds} - I_{th})] + 1}. \tag{10}$$

Equation (10) can be used to describe the high-level noise characteristics of SiC MOS-FETs in the subthreshold region, where $I_{th}$ represents the drain current under the condition of $V_{gs} - V_{th} = 0$. $a$ and $b$ are the adjustable parameters to control the influence of (10) on the above-threshold region.

Finally, by combining Equations (1) and (10), a unified LFN model can be obtained

$$\frac{S_{Ids}}{I_{ds}^2} = \frac{S_{Ids,abv}}{I_{ds}^2} + \frac{S_{Ids,sub}}{I_{ds}^2}. \tag{11}$$

The final model can characterize the noise properties of the device in multiple operation regions, especially for the high-level noise behavior of SiC MOSFETs in the subthreshold region.

## 3. Calculation of Trap State Density

The subthreshold characteristics of SiC MOSFETs are strongly affected by the trap states. In the subthreshold region, the trap state density of the device is related to the subthreshold swing ($S$) and the subthreshold ideality factor ($\eta$). $S$ is denoted as [26,27]

$$S \equiv \ln 10 \frac{dV_{gs}}{d(\ln I_{ds})} = \ln 10 \frac{kT}{q} \eta \approx \frac{kT}{q} \ln 10 \left( 1 + \frac{C_s}{C_{ox}} \right), \tag{12}$$

where $C_s$ is the effective capacitance considering the trap state density in the bandgap, which can be derived by Poisson's equation as follows:

$$\nabla^2 \varphi = \frac{q}{\varepsilon_{SiC}} n_{TR}, \tag{13}$$

where $\varphi$ is the electric potential, and $\varepsilon_{SiC}$ is the dielectric constant of SiC. Note that in the subthreshold region, since most of the induced charge is trapped in deep or tail states, the above Poisson's equation only considers the trapped charge and ignores the free electron [28]. The total energy distribution of bulk traps, i.e., the sum of the exponential distributions of the deep and tail states, can be approximated by a single exponential function as the effective trap state density. Therefore, the density of ionized traps is given as a function potential by [29]

$$n_{TR} = N_{DA} \theta_t \exp\left( \frac{E_{F0} - E_C}{kT_{DA}} \right), \tag{14}$$

where $N_{DA}$ represents the density of the trap states, and $T_{DA}$ is the characteristic temperature. Herein, $E_{F0}$ is the Fermi level in the neutral SiC layer, $E_C$ is the conduction band energy, and $\theta_t = (\pi T / T_{DA}) / \sin(\pi T / T_{DA})$ [30].

Substituting Equation (14) into Equation (13), solving the surface electric field in combination with the relation $2(d\varphi/dx)(d^2\varphi/dx^2) = (d/dx)(d\varphi/dx)^2$, and applying Gauss's law [31], we have

$$Q_s = \sqrt{2q\varepsilon_{SiC}}\sqrt{n_{TR}\frac{kT_{DA}}{q}\left[\exp\left(\frac{\varphi_s}{kT_{DA}/q}\right) - 1\right]}, \tag{15}$$

where $\varphi_s$ is the surface potential. Finally, $C_s$ is defined as the first derivative of $Q_s$ with respect to the surface potential [26], i.e.,

$$C_s = \sqrt{\frac{q\varepsilon_{SiC}}{2}}\left\{n_{TR}\frac{kT_{DA}}{q}\left[\exp\left(\frac{\varphi_s}{kT_{DA}/q}\right) - 1\right]\right\}^{-0.5}\left[n_{TR}\exp\left(\frac{\varphi_s}{kT_{DA}/q}\right)\right]. \tag{16}$$

As a consequence, using Equations (12) and (16), the trap state density of SiC MOSFETs can be extracted without using any adjustable parameters, and it can be concluded that $S$ will increase with the trap state density.

## 4. Results and Discussion

The proposed model is validated based on the existing experimental data on the LFN of the two 4H-SiC MOSFETs, i.e., Device 1 [1] and Device 2 [32]. The parameters used are summarized in Table 1. These two devices studied are both 1.2 kV class 4H-SiC MOSFETs with vertical double implant structures (VDMOSFETs), and the gate length ($L_g$) is 0.5 μm. However, they differ in gate width and oxide layer thickness, and the detailed parameters are given in [1,32]. Figure 1 shows the schematic cross-section of a single VDMOSFET elementary cell.

**Table 1.** Parameters used for model simulations in Figure 4.

| Parameters | Figure 4a (Device 1) | Figure 4b (Device 2) |
|---|---|---|
| $S_{Vfb}$ (V$^2$/Hz) | $4 \times 10^{-11}$ | $1.15 \times 10^{-12}$ |
| $V_{ds}$ (V) | 0.1 | 0.14 |
| $T$ (K) | 298 | 298 |
| $\eta$ | 3.5 | 11.35 |
| $\mu_{eff}$ (cm$^2$/Vs) | 3.3 | 0.5 |
| $V_{th}$ (V) | 5 | 2 |
| $C_{ox}$ (F/cm$^2$) | $2 \times 10^{-8}$ | $1 \times 10^{-8}$ |
| $\varepsilon_{SiC}$ (F/cm) | $8.6 \times 10^{-13}$ | $8.6 \times 10^{-13}$ |
| $f$ (Hz) | 10 | 1.22 |
| $\alpha_H$ | $2.86 \times 10^{-4}$ | $1.95 \times 10^{-4}$ |

Following the experimental results of Device 1, the frequency dependence of $S_{Ids}$ in 4H-SiC MOSFETs is displayed in Figure 2. It can be seen that the noise in both the subthreshold and above-threshold regions follows the law of $1/f^\gamma$. The value of $\gamma$ in each region is close to one (this phenomenon can also be observed in Device 2), indicating that the $1/f$ noise model derived in this work can characterize the LFN behavior of SiC devices.

Figure 3a shows the log–log plot of $S_{Ids}/I_{ds}^2$ versus $V_{th}-V_{gs}$ when Device 1 is in the subthreshold region. According to the dependence of $S_{Ids}/I_{ds}^2$ on $V_{th}-V_{gs}$ in the figure, it can be seen that the slope of the drain current noise PSD against the overdrive voltage is about $-1.3$, which shows that the LFN is produced by the combined action of carrier number fluctuation and bulk mobility fluctuation [33]. It also indicates the necessity and rationality of considering bulk defects in the subthreshold region. In addition, Figure 3b demonstrates that the slope in the log–log plot of $S_{Ids}$ against $I_{ds}$ in the subthreshold is in the range of 1–2, which also suggests that the LFN source is now associated with both the $\Delta N$ and $\Delta\mu$ mechanisms [34].

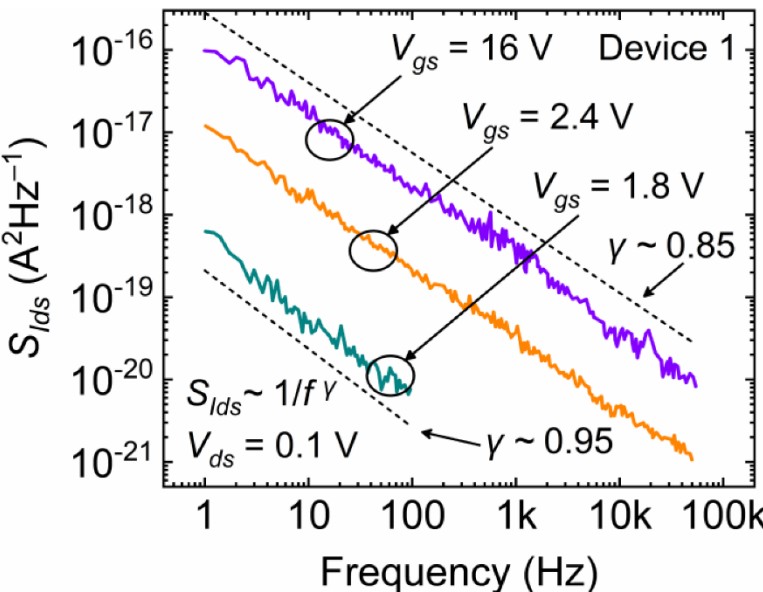

**Figure 2.** Relationship between the frequency and $S_{Ids}$, measured at different gate voltages from the subthreshold region ($V_{gs}$ = 1.8 V) to the above-threshold region ($V_{gs} \geq$ 2.4 V). The value of $\gamma$ in the subthreshold and above-threshold regions are approximately 0.95 and 0.85, respectively.

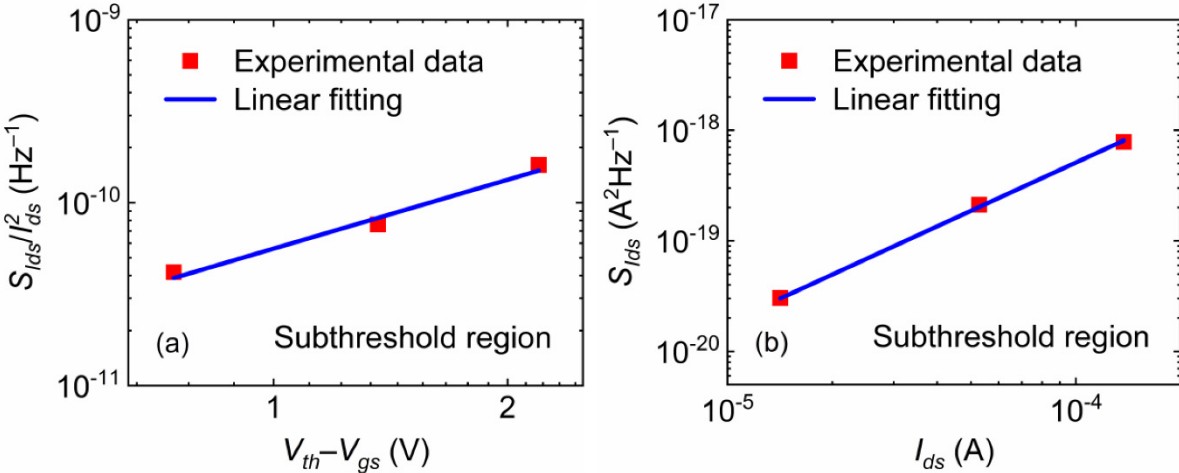

**Figure 3.** The log–log plots of (**a**) $S_{Ids}/I_{ds}^2$ versus $V_{th}-V_{gs}$ and (**b**) $S_{Ids}$ versus $I_{ds}$ in the subthreshold region.

The trap state density of the two SiC MOSFETs is extracted following the method in Section 3. $C_s$ is calculated from the extracted subthreshold swing ($S$), and then the value of $N_{DA}$ is deduced through the expression of $C_s$. For the 4H-SiC devices in Figure 4a,b, the obtained $S$ is 0.2085 V/dec and 0.6760 V/dec, respectively. Thus, the corresponding extracted $N_{DA}$ results are $1.09 \times 10^{19}$ cm$^{-3}$ eV$^{-1}$ and $4.67 \times 10^{19}$ cm$^{-3}$ eV$^{-1}$, respectively, which are reasonable as shown in [18]. In addition, it can be observed that the $S$ value of Device 2 is larger than that of Device 1, indicating that Device 2 has a higher trap state density. Moreover, from Table 1, we can see that the effective mobility ($\mu_{eff}$) of Device 2 is lower than that of the other device. The above analysis shows that the bulk defects of the device will affect its mobility in the subthreshold region and further cause LFN. Therefore, the contribution of mobility fluctuation to LFN in the subthreshold region cannot be ignored.

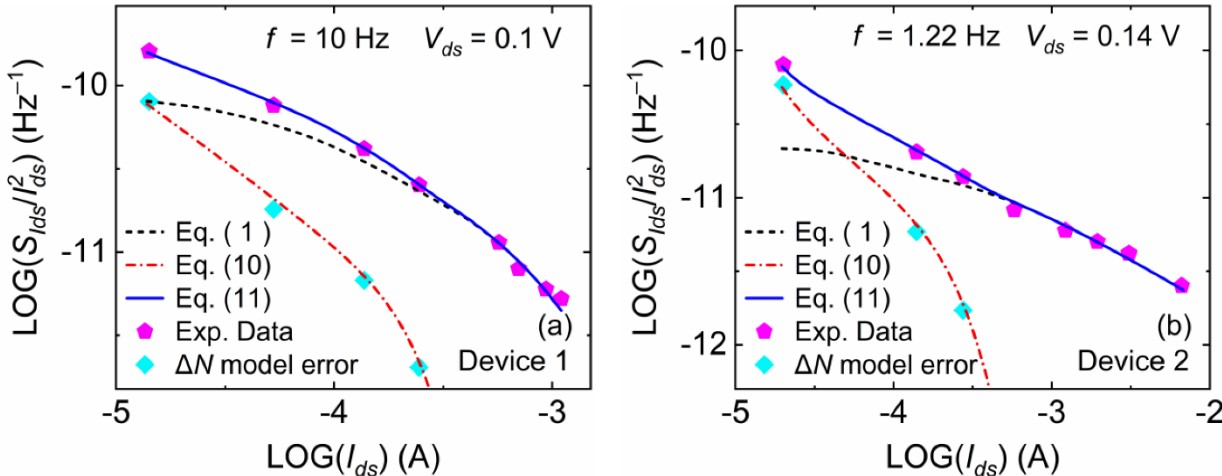

**Figure 4.** Plots of $\mathrm{LOG}\left(S_{Ids}/I_{ds}^2\right)$ versus $\mathrm{LOG}(I_{ds})$ for (**a**) Device 1 and (**b**) Device 2. The experimental data in both 4H-SiC MOSFETs are represented by symbols, and the model results are represented by lines.

Figure 4 shows the fitting results of the unified model (11) on 4H-SiC devices. It can be seen that only the $\Delta N$ model (1) can describe the LFN behavior in the above-threshold region but not for the subthreshold region. When the devices enter the subthreshold region, significant noise occurs and the measured results of LFN will be higher than the prediction of the $\Delta N$ model. This is because the dependence of $S_{Ids}/I_{ds}^2$ on $I_{ds}$ usually tends to be saturated for c-Si MOSFETs with low current in the subthreshold region, and the $\Delta N$ model is improved to accurately describe this LFN property. However, for SiC MOSFETs, the dependence of $S_{Ids}/I_{ds}^2$ on $I_{ds}$ maintains a relatively fixed law (i.e., $S_{Ids}/I_{ds}^2 \propto 1/I_{ds}^{0.5}$) from the subthreshold to the above-threshold region [1]. Hence, the phenomenon of significant subthreshold noise will appear when the classic $\Delta N$ model is directly applied.

Furthermore, the generation of high-level noise is also related to the fact that the $\Delta N$ theory only takes into account the interfacial oxide traps but not the bulk defects of SiC [23], and since the bulk trap state density of poly-Si TFTs is comparable to that of SiC MOSFETs, a similar high-level noise phenomenon will also occur when the LFN properties of poly-Si TFTs are directly described by the $\Delta N$ model. This suggests that the $\Delta N$ theory alone cannot well explain the characteristics of LFN; that is to say, considering only the effect of the interface traps cannot accurately characterize the subthreshold noise behavior of the device. It is also consistent with our analysis in Section 2, and therefore, it is necessary to consider the contribution of bulk defects to the noise in the subthreshold region. Considering the significant subthreshold noise through the $\Delta\mu$ model of (10) and LFN in the above-threshold region by (1), the unified noise model (11) can well fit the LFN phenomena of the two different SiC devices in Figure 4.

## 5. Conclusions

The $1/f$ noise in 4H-SiC MOSFETs is interpreted by the combination of mobility fluctuation and carrier number fluctuation models. The origin of the observed high-level subthreshold noise is identified as SiC bulk defects and oxide traps around the interface. In addition, the extraction of trap density is easily derived from the subthreshold swing. Ultimately, experimental noise data are analyzed and the proposed unified model describes adequately the drain current dependence of the drain current noise over a wide operation region. This improved model can be served as a good understanding of the noise characteristics of SiC MOSFET technologies.

**Author Contributions:** Conceptualization, Y.L. and W.D.; methodology, Y.L. and W.Y.; investigation, X.X.; data curation, W.Y.; writing—original draft preparation, W.Y.; writing—review and editing, Y.L.; supervision, W.D., Y.L. and W.Y. contributed equally to this work. All authors have read and agreed to the published version of the manuscript.

**Funding:** This research was funded by the Key-Area Research & Development Program of Guangdong Province under Grants 2020B010170002 and 2020B010173001, and the Science & Technology Program of Guangdong under Grant 2019A1515012127.

**Conflicts of Interest:** The authors declare no conflict of interest.

**Abbreviations**

| | |
|---|---|
| LFN | Low-Frequency Noise |
| SiC | Silicon Carbide |
| MOSFETs | Metal-Oxide-Semiconductor Field-Effect Transistors |
| c-Si | Monocrystalline Silicon |
| Poly-Si | Polycrystalline Silicon |
| TFTs | Thin-Film Transistors |
| PSD | Power Spectral Density |
| VDM | Vertical Double Implant |

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
