# Peer review of "Low-Frequency Noise Modeling of 4H-SiC Metal-Oxide-Semiconductor Field-Effect Transistors"

_electronics, doi:10.3390/electronics11193050_

Round 1
Reviewer 1 Report
Paper is well organized, well referenced and clearly written.
Minor recommendations concerning Figure 2: (1) list value of gamma parameter for line drawn in the figure caption; (2) move figure to be after the table so it is closer to where it is referenced in the text.
Author Response
Response to Reviewer1 Comments (electronics-1924078)
Reviewer 1
Paper is well organized, well referenced and clearly written.
Point 1:
Minor recommendations concerning Figure 2: (1) list value of gamma parameter for line drawn in the figure caption; (2) move figure to be after the table so it is closer to where it is referenced in the text.
Response 1:
Thank you very much for your helpful comments. In the revised manuscript, we have modified the caption of Figure 2 by adding the γ value of the line drawn. Also, we have moved Figure 2 after Table I.
Corresponding change in manuscript: Yes.
Location of Change: Figure 2 in Section 4 on page 6.

Reviewer 2 Report
In this manuscript the authors presented an improved low-frequency noise (LFN) model for 4H-silicon carbide metal-oxide-semi-9 conductor field-effect transistors, which is based on 10 the mobility fluctuation (Δμ) theory for excess subthreshold noise explanation, as well as the carrier 11 number fluctuation (ΔN) theory due to the oxide traps near the gate-insulator/active-layer interface.
Although this Review seems interesting, but before publishing, I urge the authors to make the following improvements considering my primary recommendations.
1- It will be good to start the abstract of the manuscript with a precise introductory sentence and then rest of the explanation about this work. I suggest the authors to revise the abstract and introduction and make its story line clearer and more directional.
2- The author tried to explain the PSD of flat-band voltage in equation 2. Could you explain it in detail that how did you calculate the oxide capacitance Cox for the insulating layer?
3- The recently reported papers should be cited in the introduction section (Nanomaterials (Basel). 2019 Mar; 9(3): 422.) & ( https://doi.org/10.1002/adfm.202204781).
4- The English language need to be improved.
Remarks: Publish after Minor revision
Author Response
Response to Reviewer2 Comments (electronics-1924078)
Reviewer 2
In this manuscript the authors presented an improved low-frequency noise (LFN) model for 4H-silicon carbide metal-oxide-semiconductor field-effect transistors, which is based on the mobility fluctuation (Δμ) theory for excess subthreshold noise explanation, as well as the carrier number fluctuation (ΔN) theory due to the oxide traps near the gate-insulator/active-layer interface.
Although this Review seems interesting, but before publishing, I urge the authors to make the following improvements considering my primary recommendations.
Remarks: Publish after Minor revision
Point 1:
It will be good to start the abstract of the manuscript with a precise introductory sentence and then rest of the explanation about this work. I suggest the authors to revise the abstract and introduction and make its story line clearer and more directional.
Response 1:
According to your valuable suggestion, we have revised the Abstract and Introduction of the manuscript to make its structure more logical.
Corresponding change in manuscript: Yes.
Location of Change: Abstract and Introduction.
Point 2:
The author tried to explain the PSD of flat-band voltage in equation 2. Could you explain it in detail that how did you calculate the oxide capacitance Cox for the insulating layer?
Response 2:
For Device 1 and Device 2 in Section 4, no explicit Cox value is given in [1] and [32]. In this paper, by fitting the I-V transfer characteristic curve by Equation (8), the corresponding Cox value can be extracted.
Corresponding change in manuscript: No.
Point 3:
The recently reported papers should be cited in the introduction section (Nanomaterials (Basel). 2019 Mar; 9(3): 422.) & (https://doi.org/10.1002/adfm.202204781).
Response 3:
Thank you very much for your advice. In the revised manuscript, we have included these papers in the Introduction section.
Corresponding change in manuscript: Yes.
Location of Change: Introduction and Reference.
Point 4:
The English language need to be improved.
Response 4:
Thanks for your comment. Since English is not our mother tongue, we may not be doing enough in some aspects of expression. In the revised manuscript, we have improved some expressions.
Corresponding change in manuscript: Yes.
Location of Change: Full article.

Reviewer 3 Report
An improved low-frequency noise (LFN) model for 4H-silicon carbide metal-oxide-semiconductor field-effect transistors (4H-SiC MOSFETs) is presented in this paper, which is based on the mobility fluctuation (Δμ) theory for excess subthreshold noise explanation, as well as the carrier number fluctuation (ΔN) theory due to the oxide traps near the gate-insulator/active-layer interface.
Comments:
Please improve the introduction.
Please explain the effect of Lg on noise.
Please consider the length of channel in the analysis.
Author Response
Response to Reviewer3 Comments (electronics-1924078)
Reviewer 3
An improved low-frequency noise (LFN) model for 4H-silicon carbide metal-oxide-semiconductor field-effect transistors (4H-SiC MOSFETs) is presented in this paper, which is based on the mobility fluctuation (Δμ) theory for excess subthreshold noise explanation, as well as the carrier number fluctuation (ΔN) theory due to the oxide traps near the gate-insulator/active-layer interface.
Point 1:
Please improve the introduction.
Response 1:
Thank you for your valuable suggestion, we have made improvements to the Introduction section of the manuscript.
Corresponding change in manuscript: Yes.
Location of Change: Introduction.
Point 2:
Please explain the effect of Lg on noise.
Response 2:
The increase of the channel length (Lg) will reduce the low frequency noise (LFN), which can be explained by the ΔN model and the Δμ model together. For the ΔN model, it can be seen from the definition of SVfb (Equation (2)) that when Lg increases, SVfb will decrease, and then the noise spectral density will reduce. For the Δμ model, the above conclusion can also be obtained from Equation (5). This conclusion is applicable to 4H-SiC MOSFETs, poly-Si MOSFETs and a-IZO TFTs, which can be found in [1], [R1], and [R2], respectively.
Corresponding change in manuscript: Yes.
Location of Change: Below Equation (2) in Section 2, we have modified the original content to “W and Lg are the width and length of the gate as shown in Figure 1, it can be seen from Equation (2) that when Lg increases, SVfb will decrease, and then the noise PSD in Equation (1) will reduce.” And below Equation (5), we have added “Like the ΔN model, the noise PSD described by the Δμ model will also reduce as Lg increases.”
[R1] Dimitriadis, C.A.; Brini, J.; Kamarinos, G.; Gueorguiev, V.K.; Ivanov, T.E. Conduction and low-frequency noise in high temperature processed polycrystalline silicon thin film transistors. J. Appl. Phys. 1998, 83, 1469-1475. [DOI: 10.1063/1.366852]
[R2] Liu, Y.; He, H.; Chen, R.; En, Y.F.; Li, B.; Chen, Y.Q. Analysis and simulation of low-frequency noise in indium-zinc-oxide thin-film transistors. IEEE J. Electron Devi. 2018, 6, 271-279. [DOI: 10.1109/JEDS.2018.2800049]
Point 3:
Please consider the length of channel in the analysis.
Response 3:
We agree that it’s better to analyze noise data with various channel lengths when compared with Device 1 and Device 2 originated from [1] and [32]. However, there is no other same devices but different channel lengths in [1] and [32], and therefore, it is difficult to complete this analysis. Nevertheless, as we described in the reply of Comment 2, our noise model can explain the effect of Lg on noise.
Corresponding change in manuscript: No.
